# Changes in Nutrition Impact Symptoms, Nutritional and Functional Status during Head and Neck Cancer Treatment

**DOI:** 10.3390/nu12051225

**Published:** 2020-04-26

**Authors:** May Kay Neoh, Zalina Abu Zaid, Zulfitri Azuan Mat Daud, Nor Baizura Md. Yusop, Zuriati Ibrahim, Zuwariah Abdul Rahman, Norshariza Jamhuri

**Affiliations:** 1Department of Nutrition and Dietetics, Faculty of Medicine and Health Sciences, Universiti Putra Malaysia, Seri Kembangan 43400, Selangor, Malaysia; dtneoh@nci.gov.my (M.K.N.); zulfitri@upm.edu.my (Z.A.M.D.); norbaizura@upm.edu.my (N.B.M.Y.); zuriatiib@upm.edu.my (Z.I.); dtshariza@nci.gov.my (N.J.); 2Department of Dietetic and Food Service, National Cancer Institute, Ministry of Health, 4, Jalan P7, Presint 7, Putrajaya 62250, Malaysia; dtzuwariah@nci.gov.my

**Keywords:** head and neck cancer, nutritional status, malnutrition, nutrition impact symptoms, dietary intake, oral nutritional supplements

## Abstract

Background: The purpose of this study is to evaluate changes in nutrition impact symptoms (NIS) and nutritional and functional status that occur throughout radiotherapy in head and neck cancer (HNC) patients. Methods: A prospective observational study of HNC inpatients who underwent radiotherapy with or without chemotherapy were recruited to participate. Fifty patients were followed for the periods before, in the middle and at the end of radiotherapy. Nutritional parameters were collected throughout radiotherapy. Results: According to Patient-Generated Subjective Global Assessment (PG-SGA), there was an increase from a baseline of 56% malnourished HNC patients to 100% malnourished with mean weight loss of 4.53 ± 0.41kg (7.39%) at the end of radiotherapy. Nutritional parameters such as muscle mass, fat mass, body mass index, dietary energy and protein intake decrease significantly (*p* < 0.0001) while NIS score, energy and protein intake from oral nutritional supplements (ONS) increased significantly (*p* < 0.0001). Hand grip strength did not differ significantly. All HNC patients experienced taste changes and dry mouth that required ONS at the end of treatment. ONS compliance affected the percentage of weight loss (*p* = 0.013). Conclusions: The intensive nutritional care time point was the middle of RT. The PG-SGA and NIS checklist are useful for monitoring nutrition for HNC patients.

## 1. Introduction

Head and neck cancer (HNC) involves malignancies of the oral and nasal cavities, sinuses, salivary glands, pharynx, larynx, and lymph nodes in the neck. It is the sixth most frequently observed type of cancer worldwide [1]. HNC patients are often malnourished at the time of diagnosis. Prior to the beginning of treatment and during treatment, this is due to the catabolic state induced by the tumor and the side effects from treatment, respectively. Unintentional weight loss is common in HNC, mainly because of the tumor location and related symptoms that interfere with dietary intake. The prevalence of malnutrition in HNC patients at diagnosis ranges from 42% to 77% and worsen throughout the treatment [2,3], with rises up to 88% [4].

The current treatment of advanced HNC requires multimodal therapy. Surgery, radiotherapy (RT), concurrent chemotherapy and radiotherapy (CCRT) have become standards of care for HNC patients. Treatment has improved tumor control [5,6] but brings about various side effects. The common side effects of RT with or without chemotherapy in HNC patients are mucositis, dysphagia, xerostomia, altered taste and smell, and chewing and swallowing difficulty [4,7]. These nutrition impact symptoms (NIS) induce pain and an inflammatory response, which limits energy intake and increases the stress response, resulting in weight loss [8] during treatment.

Weight loss among HNC patients during treatment is clinically relevant as it contributes to ineffective treatment responses and an impaired functional performance status, leading to a reduced quality of life and a significantly lower survival rate [5,6,9]. Critical weight loss is defined when body weight loss is >5% [10]. Langius et al. (2013) found out that patients who experience weight loss >5% during RT are independently associated with a 1.7 times higher risk of dying from HNC [10]. The cut-off points for critical weight loss during RT were based on the international consensus statement of the Academy of Nutrition and Dietetics and the American Society for Parenteral and Enteral Nutrition [11]. A HNC patient should be weighed routinely, since any weight loss >5% (compared to their usual weight or healthy weight over six months) is a factor contributing to a poor prognosis [12].

The duration of RT for HNC patients is typically set around six to seven weeks and the dosage of RT is determined by the oncologist. This period is the critical period for nutrition assessment, intervention, and monitoring due to rapid changes of NIS and dietary intake. In the HNC population, symptom assessment is often adopted in an effort to restore dietary intake and reduce unintentional weight loss, especially during treatment [13]. Patients will develop NIS at the beginning or middle of their treatment that requires different intensities of nutrition intervention. The European Society for Clinical Nutrition and Metabolism (ESPEN) recommends special attention to RT for HNC, where an adequate nutritional intake should be ensured primarily by individualized nutritional counselling and/or with use of oral nutritional supplements (ONS), in order to avoid nutritional deterioration [14]. The majority of HNC patients receiving RT are recommended for ONS as it has been shown that fat mass and fat free mass remain unaltered in compliant patients during treatment [12], but the timing for initiation has yet to be identified. 

Due to the lack of studies that have investigated the progression of NIS, nutrition status, and energy intake from ONS throughout RT treatment, it is harder to identify patients in need of intensive nutritional intervention. Many of the studies have emphasized weight loss and energy intake for the pre-treatment and post-treatment periods, mainly focusing on the effect of treatment [13,15]. However, more knowledge is required to study the pattern of changes of diet, ONS energy and protein intake, and risk factors for critical weight loss during RT treatment.

This prospective observational study in HNC patients who were treated with standard treatment allows us to determine changes in NIS and nutritional and functional status among HNC patients during RT. Therefore, this study aims to determine the prevalence and magnitude of weight changes in HNC patients and the relationship with the changes of NIS and nutritional and functional status during RT treatment. 

## 2. Materials and Methods 

### 2.1. Study Design and Setting

This prospective observational study was conducted among adult HNC in patients who received RT or CCRT from March until December in 2018 at the National Cancer Institute (NCI), Putrajaya, Malaysia. This study used a consecutive technique to recruit every HNC patient who was admitted to receive RT or CCRT at NCI, Putrajaya, based on the inclusion criteria and their informed consent. The RT patients received a RT dosage between 60 to 70 Gy in daily factions of 2.0 Gy within 7 weeks, while CCRT patients received additional weekly cisplatin or carboplatin during the 7 weeks of RT. The inclusion criteria were HNC patients who were admitted to a ward having RT within 7 weeks with or without chemotherapy for curative treatment intentions and being aged 18 years or older. Participants were on a 100% oral intake at the time of the study, and no patient used any form of enteral tube feeding or total parenteral nutrition. Patients were excluded from this study if they were involved in another research project and ongoing artificial nutrition (enteral/parenteral) before RT or CCRT. There were three measurement points in this study to assess NIS, and nutritional status and functional status, included at the baseline (week 1), middle (week 4) and end of the RT (week 7).

### 2.2. Measures

#### 2.2.1. Socioeconomic-Demographic and Clinical Characteristics

The socioeconomic data included the age, gender, and ethnicity of patients. For the clinical characteristics including co-morbidities, tumor location and stage, type of treatment, and duration and dosage of RT, these were obtained from the computerized hospital information system (HIS).

#### 2.2.2. Nutritional Status

Malnutrition Status

The malnutrition statuses of patients were determined using the scored Patient-Generated Subjective Global Assessment (PG-SGA). The PG-SGA is a global rating and scoring nutritional assessment tool that is specialized for cancer patients [16]. Patients are subjectively categorized as well-nourished (PG-SGA category A), moderately or suspected of being malnourished (PGSGA category B) or severely malnourished (PG-SGA category C) upon completion of the assessment. The scored PG-SGA is a further development of the subjective global assessment (SGA) concept that incorporates a numerical score. A high score indicates a lower nutritional status that requires nutrition intervention. Scores with 0 to 1 point require no intervention, health education for 2 to 3 points, dietetic intervention for 4 to 8 points, and nutrition support for >9 points. 

Anthropometric Measurements 

The anthropometric measurements used in this study include body weight, height, and body composition. Body height was measured using a stadiometer (Seca 222, Medical Scales & Measuring Systems Seca, United Kingdom). Measurements of body weight and body composition were assessed with a calibrated Tanita total body composition analyzer (model SC 300) which can provide body weight in kg (up to 0.1 kg), fat percentage (up to 0.1%), and total muscle mass (up to 0.1 kg). The subjects were required to be bare foot and stand upright and front facing during measurement. The subjects were requested to have minimal clothing, empty pockets, and stand upright while barefoot on the metal plate of the scale. 

Body mass index (BMI) was calculated as the actual body weight/height^2^ in (kg/m^2^). BMI was then classified as either underweight (BMI <18.5 kg/m^2^), normal (BMI 18.5–24.9 kg/m^2^), overweight (BMI 25–29.5 kg/m^2^) or obese (BMI >30.5 kg/m^2^) [17]. Percentage weight loss was calculated as (normal body weight − actual body weight)/(normal body weight) × 100. Normal body weight was defined as the body weight 1 month before treatment and was retrieved from medical records. Actual body weight was assessed at the beginning of treatment. 

Dietary Intake and Oral Nutritional Supplements (ONS) Intake

Dietary intake was measured through a 24 h dietary recall. Food and beverages consumed in the last 24 h, starting from the last midnight and finishing at midnight, were identified by the 24 h dietary recall. The household serving intake, and subsequently the gram intake of food, was collected for every meal to estimate energy and macronutrient intake. Household portion sizes were used to calculate the grams of food that were consumed. For this purpose, household cups and spoons were applied. The intake of energy and macronutrients was determined.

The Nutritionist Pro software was used to analyze information about the macronutrient intake amount (in grams) and total energy intake (kcal) by entering meal recipes with the exact gram intake of all food items. The software calculates the nutrition facts of the whole foods taken in a day from recall. Data on the total energy and protein intake were recorded to compare with the energy requirements of patients. Total energy and protein intake included energy and protein from the diet and ONS, while dietary intake and protein intake included energy and protein contributed from diet alone. ONS energy and protein intake was evaluated from ONS intake during treatment.

Nutrition counselling was given to all patients at diagnosis. An oncology dietitian adapted their diet to improve their nutritional intake, especially emphasizing protein intake and the fractionation of intakes throughout the day. Only when nutritional requirements were not met with dietetic intervention was nutritional supplementation then prescribed according to individual needs.

The number of nutrition supplements cans consumed by each subject was recorded as one of the methods to measure the compliance to nutrition support. Weekly ONS compliance was assessed through intake of the ONS, starting from week 2 after dietary intervention. There was a total of 6 (week 2 until end of treatment) compliance measurements throughout treatment. If the patient consumed ≥75% of the recommended ONS energy and protein, he/she was considered as a “compliant patient” while those who consumed <75% of the recommended ONS energy and protein were considered to be “noncompliant patients” [12].

Total energy and protein intake were compared to the energy and protein requirements of patients. According to the Medical Nutrition Therapy Guidelines for Cancer in Adults, Ministry of Health (MOH) [18] and ESPEN 2017 guidelines, the recommended energy and protein requirements for hyper-catabolic patients are 30–35 kcal/kg of body weight and 1.2–1.6 g/kg of protein [14]. Hyper-catabolism is defined as any cancer patient undergoing treatment here. Energy requirements and protein requirements are based on the actual body weight, unless the patient was overweight (BMI of >25 kg/m^2^), where the adjusted body weight was used instead in that case. The adjusted body weight was calculated using the following equation: Adjusted body weight = (Ideal body weight (IBW) + ((actual body weight − IBW) × 25%), whereby IBW is the patient’s corresponding weight at a BMI of 25 kg/m^2^ [19].

#### 2.2.3. Nutrition Impact Symptoms (NIS)

The NIS were measured with the Head and Neck Symptoms Checklist^©^ (HNSC^©^). This instrument aids in the early identification of symptoms that place HNC patients at risk of reductions in dietary intake, weight, and functional performance. This checklist includes 12 of the symptoms included on the PG-SGA (including pain, dry mouth, loss of appetite, constipation, feeling full, diarrhea, sore mouth, nausea, altered smell, vomiting, difficulty swallowing, and taste changes) plus five additional symptoms (lack of energy, depression, difficulty chewing, thick saliva, and anxiety) not included on the PG-SGA but reported in the literature as being associated with reduced dietary intake [13,20]. The HNSC^©^ also provides space for patients to record any additional NIS interfering with eating. Patients were asked to rate the severity of each symptom and the degree to which it interfered with eating (dietary intake) using a five-point Likert scale ranging from “1, not at all” to “5, a lot” [13]. All 17 symptom scores in the checklist were added together to make a total symptom score which varies from 17 (no symptoms) to 85 (highest score of 5 for every symptom in the list) [8]. If acute symptoms scores were missing for only one week for a particular subject, the scores from the previous week were used [21].

#### 2.2.4. Functional Status

Functional status was measured by handgrip strength. The non-dominant hand was measured by using a Jamar hand dynamometer (Fred Sammons Inc, Burr Ridge, IL, USA).

### 2.3. Power Calculation

Minimum sample sizes were calculated as required for a paired t-test using the formula *n* = σ^2^ (Z_1_ − ∝/2 + Z_1_ − β)^2^/(μ_1_ − μ_2_)^2^ [22], where n is the number of patients, Z is the level of confidence, α is alpha (0.05), β is beta (0.2), μ_1_ is mean in time 1 and μ_2_ is mean in time 2, using body weights previously published by another study [13]. Hence, a minimum sample size of 32 samples is required to be able to reject the null hypothesis with a probability (power) of 0.8. The type I error probability associated with this test of this null hypothesis is 0.05. With an additional of 20% dropout rate, the required sample size is 40 samples.

### 2.4. Ethical Approval

This study was registered with The National Medical Research Registry (NMRR ID 17-2647-37667). Ethical approval for the study was obtained from the Medical Research Ethics Committee of the Faculty of Medicine & Health Sciences, Universiti Putra Malaysia and the Medical Research Ethics Committee (MREC), Ministry of Health Malaysia. Permission to conduct the study was obtained from the director at the NCI, Putrajaya, Malaysia.

### 2.5. Statistical Analysis

All statistical analyses were performed using the Statistical Package for the Social Sciences (SPSS) for Windows, version 23 (SPSS Inc, Chicago, IL, USA). Data were checked for normality via the Shapiro-Wilk test. All data were normally distributed as indicated by *p* > 0.05 unless otherwise stated. If the data were not normally distributed, analyses were carried out on natural logarithm of the values to improve the symmetry and homoscedasticity of the distribution. 

The descriptive statistics here include percentages, means and standard deviations, used to describe the demographic data, clinical characteristics, nutritional status, anthropometric data, biochemical data, nutrition impact symptoms, and dietary intake. The values from different groups were compared by using an independent t-test. For non-normally distributed or ordinal data, the Mann-Whitney U-test was carried out to test the differences between groups and Spearman’s rho test was used to evaluate the association between two numerical variables. A chi-square test was used to see the significant differences between groups for categorical data. Changes of fat mass, total energy and protein intake, dietary energy and protein intake, ONS energy and protein intake over time were analyzed by the Friedman test. 

Body weight, muscle mass, NIS score and handgrip strength are expressed as the mean ± SD, and changes over time were analyzed by the general linear model with repeated measures. In the case of deviation from sphericity, a Greenhouse Geisser correction for degrees of freedom was used. The relationship of continuous data were analyzed with Pearson’s correlation, while categorical data was analyzed with Spearman test. For the purpose of this analysis, patients were classified as either well-nourished (PG-SGA A) or malnourished (PG-SGA B and C)**.** A statistical probability of *p* < 0.05 was considered as significant.

## 3. Results

### 3.1. Baseline Characteristics

Fifty-four patients consented to participate (see Figure 1) for this study. A total of four patients were excluded with two patients who did not meet the study’s criteria and another two who did not complete treatment. Fifty subjects were able to complete RT treatment as inpatients and be followed up for data collection until the end of treatment. 

Body composition measurements were performed with 49 patients, as there was an error with a patient’s data during the Tanita bioelectrical impedance analysis due to a low-fat percentage with an underweight condition. In all other measurements, the data of 50 patients were used in the analysis.

The study results show that there were more male than female patients with HNC (78% versus 22%) and the mean age of the population was 57 years (SD of two) with a range of 21–78 years old. There were twenty-one (42%) Malay, nineteen (38%) Chinese, and ten (20%) Indian patients, respectively. More than half of the HNC (52%) subjects in this study had nasopharynx cancer and 84% were in an advanced stage of the tumor. In addition, thirty-three (60%) received CCRT while seventeen (34%) received RT only. All subjects received a total of 60 Gy and above 30 fractions of radiation dosage (Table 1).

### 3.2. Changes in Nutritional Status, Nutrition Impact Symptoms (NIS) and Functional Status 

#### 3.2.1. PGSGA

The pre-treatment prevalence of malnutrition was 56% (28 of 50). Only one third of the malnourished patients had received dietary intervention before treatment. At the end of treatment, all subjects were malnourished, where 32% (16) were moderately malnourished and 68% (34) were severely malnourished (Table 2). There was an increase of severely malnourished HNC patients from 20% at the baseline to 68% at the end of treatment. All HNC patients required critical intervention at the end of treatment. At the end of treatment, 68% HNC patients had >5% weight loss with 30% of them experiencing >10% severe weight loss. 

#### 3.2.2. Anthropometric Data

The mean pre-treatment body weight significantly declined from 60.24 ± 14.73 kg to 57.95 ± 13.92 kg at the middle of treatment and further reduced to 55.71± 13.62 kg at the end of treatment (*p* < 0.001; mean decline of 4.53 ± 2.87 kg). The muscle mass at end of treatment was 41.3 ± 8.13 kg, which was declined from the baseline of 43.03 ± 8.12 kg (*p* < 0.001, mean decline of 1.73 ± 2.17 kg) (Table 3). Repeated measures of analysis of variance showed that marked decreases were noted in anthropometric data which included the mean body weight, muscle mass and fat mass from baseline to mid-treatment and from mid-treatment to post-treatment. At the end of treatment, 42% of patients had a normal weight, 12% were overweight, 8% were obese, followed by increases of being underweight from 24% at baseline to 38% at the end of treatment (Table 3). Patients lost 4.53 ± 0.41 kg (95% CI, 3.72 to 5.35 kg) of body weight, which corresponds to an average of 7.39% body weight loss (Table 2). 

#### 3.2.3. NIS

About 80% HNC patients had NIS before the start of treatment, with only 20% of patients experiencing no NIS (Table 2). At the end of treatment, all subjects experienced NIS (Table 2). The mean symptoms score increased from 21.78 ± 4.59 to 39.68 ± 9.54 at the middle of treatment and up to 48.34 ± 8.79 at the end of treatment (Table 3). At the baseline, 52% of HNC patients had the symptom of difficulty chewing (52%). At the middle of treatment, all HNC patients experienced almost all NIS, except diarrhea, depression, and vomiting. All 50 HNC patients experienced taste changes and a dry mouth at the end of treatment. More than 80% of them experienced difficulty swallowing, difficulty chewing, a loss of appetite, pain, lack of energy, and a sore mouth (Table 4).

#### 3.2.4. Dietary and ONS Intake

At the end of treatment, normal and soft diet reduced to 42% while liquid diet increased to 46% and tube feeding increased to 12% (Table 2). The total energy and protein intake of HNC patients at baseline was 1412 ± 589 kcal/day and 63 ± 30 g/day respectively. Both results had increased to 1554 ± 482 kcal/day and 66 ± 21 g/day at the end of treatment (Table 3). 

As shown in Table 3, total energy and total protein did not change significantly over time. However, there were significant changes for dietary energy, dietary protein, ONS energy and ONS protein over time. Dietary energy intake and protein intake decreased significantly throughout the treatment while ONS energy and protein intake increased significantly throughout the treatment. All HNC subjects required ONS during RT treatment. The provision of energy and protein from ONS was 1142 ± 572 kcal/day and 49.99 ± 24.42 g/day at the end of treatment (Table 3). This means that more than 70% of the total energy and protein intake was from ONS among HNC patients at the end of treatment.

#### 3.2.5. Handgrip Strength

There was a decreasing trend of non-dominant handgrip strength from 25.17 ± 7.46 kg at the beginning, 25.06 ± 8.26 kg in the middle, to 24.45 ± 7.97 kg at end of treatment (Table 3). Handgrip strength did not significantly change over time (*p* = 0.483) (Table 3). 

#### 3.2.6. ONS Compliance

There were 37 patients (74%) who consumed sufficient (≥75% of their requirements) ONS energy and protein and were considered to be “compliant”. Consumption did not meet the criteria of ≥75% of the recommended ONS energy and protein intake in 26% (*n* = 13) of patients, who were therefore considered to be “non-compliant” (Table 2). There was a significant difference in the percentage of end treatment weight loss between the ONS compliant group (6.55 ± 3.65%) versus the non-compliant group (9.75 ± 4.4%), with a t-value of −2.570, where *p* = 0.013. (Table 5). The total energy and total protein intakes for the ONS compliant group and ONS non-compliant group were 1653 ± 461 kcal/day and 71 ± 19 g/day, and 1273 ± 439 kcal/day and 51 ± 18 g/day, respectively. Significant differences of total energy and total protein were found in the intake between the ONS compliant group and the ONS non-compliant group (*p* < 0.05) (Table 5). The ONS non-compliant group experienced significant higher muscle mass loss (2.78 ± 2.61 kg) compared to the compliant group (Table 5). 

### 3.3. Correlations between Weight Loss and Others Nutrition Parameters

The PG-SGA score at the end of treatment had a significantly strong positive relationship with weight loss (*r* = 0.6, *p* < 0.001) (Table 6). There was a significant positive relationship between the percentage of weight loss at the end of treatment with the total symptoms score (*r* = 0.363, *p* = 0.01) (Table 6). The higher total symptoms score among HNC subjects, the higher percentage of weight loss at the end of treatment.

### 3.4. Correlation Between Total NIS Score and ONS Energy and Protein Intake

There was a significant positive relationship between the total nutrition impact symptoms score and ONS energy intake at the middle of RT (*r* = 0.429, *p* = 0.002) and a significant positive relationship between the total nutrition impact symptoms score and ONS protein intake (*r* = 0.412, *p* = 0.003) at the middle of RT (Table 7).

## 4. Discussion

All HNC patients were malnourished at the end of treatment according to PG-SGA in this study (Table 2). Capuano’s study showed a strong correlation between involuntary weight loss and PG-SGA, which was similar to our study, where a high PG-SGA score had a strong positive relationship with a high percentage of weight loss at the end of treatment [23]. There was a high prevalence of critical weight loss (>5%) among HNC at the end of treatment (68%) and 30% of them had weight loss greater than 10%. Patients with critical weight loss during RT had worse disease-specific survival related to poor response to treatment. There was an impaired immune response as a consequence of insufficient food intake in malnourished patients [10,21]. Weight monitoring is very important during RT as this treatment requires head masks for precise patient positioning to ensure accurate radiation delivery to target area. Any weight loss during RT will lead to inaccurate radiation delivery to the target area, which can reduce the effectiveness of treatment (i.e., treatment interruption) [24]. Weight maintenance leads to beneficial outcomes and is an appropriate aim for nutritional interventions.

The average daily energy and protein intake in this study was below the ESPEN recommended guideline before treatment [14]. Even though the average daily energy and protein intake were improved at the end of treatment compared to the baseline (Table 3), the intake was still below the recommendation of 30 kcal/kg/day and 1.2 g/kg/day, respectively [14]. This was clearly insufficient because the majority of patients continued to lose weight throughout the treatment because of inadequate energy and protein intake. According to the Prevost review study, optimal nutrition status before treatment is able to improve the effectiveness of treatment and improve treatment outcomes [25]. Malnutrition before treatment will increase the risk of infection [8] and further decrease the survival rate of patients [10]. In this study, 72% patients had a pre-treatment weight loss problem, where 12% had critical weight loss (<5%), which indicates the importance to start nutrition intervention before treatment (Table 2). 

The results of our study show a higher intake of calories and protein at the time that treatment was completed, as compared to the baseline intake. This result is comparable with those of another study [26]. This can be related with more than half of the HNC patients, who had malnutrition at the baseline of treatment due to inadequate food and nutrition knowledge. Some studies have revealed that most cancer patients, having dietary perceptions and beliefs with specific foods restrictions, result in potential malnutrition after diagnoses [27,28,29]. 

At the baseline, half of the HNC patients had the symptom of difficulty chewing, which relates to dental extraction one month prior to RT or a tumor located around the oral cavity, affecting the opening of the mouth or post-operation effects on jaw movement [8]. Neoadjuvant chemotherapy prior to RT is one of the reasons HNC patients experienced loss of appetite, dry mouth, lack of energy, thick saliva, and pain. For a patient who is having nutrition impact symptoms prior to RT, ONS initiation should be implemented as soon as beginning of the RT. If possible, HNSC© is recommended for use during neoadjuvant chemotherapy or as soon as the diagnosis is confirmed, as it is a useful tool to predict dietary intake. 

Our study reported that all HNC patients experienced multiple NIS at the end of treatment, resulting in a high NIS score (Table 4), which was a similar observation from a study Kubrak et al. (2013) [13]. Multiple NIS are more likely to reduce dietary intake, induce weight loss, and decrease the chance of survival. Post-dental extraction prior to treatment might lead to difficulty in chewing hard solid foods prior to treatment. This may worsen during treatment due to the complication of other NIS that further compromise dietary intake. A few research studies of HNC patients have reported a significant association between NIS and reduced dietary intake, weight loss, and reduced functional performance [13,14,15]. Almost 50% of our subjects chose liquid diet rather than normal texture diets due to worsening NIS in the middle of treatment and highest at the end of treatment. Foods that are cold, moist, soft, and with sauces, broths or accompanying meals with liquids are better choices for dry mouth patients. 

There was an association between weight loss and NIS score (Table 6). The higher NIS score was due to RT resulting in discomfort and difficulty with eating. Most of the patients burdened by NIS prefer to choose ONS as a meal replacement vector. Most ONS were liquids, which may be less satiating and easier to consume than solids when individuals are suffering from pain, loss of appetite or poor dentition [30]. Some patients who suffered from oral mucositis and nausea with bothersome smell problems were able to fulfill compliance well via ONS rather than foods [26]. A greater percentage of energy was derived from liquids which related to a higher NIS burden experienced during treatment. The same result was reported by Nejatinamini study [26]. About 70% of patients in this study selected ONS as a major contributor to energy intake at the end of treatment. Liquid milk has less triggers for nausea sensation and no chewing effort is required, which reduces the consumption time when compared to solid foods. ONS also contains complete balance nutrients of a sufficient amount to achieve energy and protein requirements, which encourages patients’ easy compliance to ONS rather than foods, with neither different types of texture modification nor portion size tolerance modification.

Our study revealed that the ONS non-compliant group experienced significantly higher percentage weight loss, muscle mass loss, lower total energy and protein intake compared to the compliant group (Table 5). Hopanci et al. showed a similar result in their HNC non-compliant group [14]. The ONS non-compliant group was unable to achieve the minimum energy intake of 25 kcal/kg/day and protein intake of 1 g/kg/day. This group lost almost 10% weight, which has greater risk of death according to a cohort study by Langius et al. (2013) [10]. The energy density, volume, and flavor of ONS supplied may affect the compliance rate [31]. A study from McCurdy et al. (2019) suggested intakes greater than 30 kcal/kg BW/day may be required to prevent weight loss as well as muscle mass loss [32]. The ONS compliant group was able to achieve the recommended protein intake (1.2 g/kg/day) and experience less muscle mass loss compared to the non-compliant group. 

In this study, the ONS compliant group had >5% critical weight loss despite dietary counselling and ONS support. The average ONS compliant group achieved 28 ± 9 kcal/kg, which was below 30 kcal/kg/day when compared with the current recommendations. The standard compliance rate in this study, according to Hopanci’s study, was set at ≥75% as a benchmark for compliance [12]. However, the ONS compliant group in this study still experienced critical weight loss which means that 75% could not reach the standard compliance rate for HNC patients undergoing RT. Perhaps a suggestion of a 100% compliance rate can be considered as compliance for better nutrition outcome monitoring. Therefore, the role of a dietitian in counselling with the support of ONS and the process along the treatment are very important to increase patients’ compliance towards nutrition intervention in order to effectively optimize nutrition status. 

This study has shown that body weight, muscle mass, fat mass, dietary energy and protein intake followed a decreasing trend along the treatment, while the PG-SGA score, NIS score, and ONS energy and protein intake followed an increasing trend (Table 3). Patients in the current study were unable to maintain their lean mass during head and neck cancer treatment. Instead, a significantly higher muscle mass loss was seen among HNC patients in the non-compliant group. The decrease of muscle mass in the patients with head and neck cancer in the middle of RT and at the end of RT may reveal the time points where nutritional care should to be provided to patients [24]. 

In the middle of RT, ONS energy and protein intake contributed to more than half of the energy and protein intake with the majority of patients experiencing a high score of NIS that interrupted dietary intake. The NIS score and ONS intake show a significant positive relationship at the middle of the RT. A higher NIS score requires higher ONS intake, as HNC patients suffer from multiple NIS, including a loss of appetite, dry mouth, taste changes, a sore mouth, difficulty swallowing, difficulty chewing, thick saliva and pain that interrupt eating. ONS should be initiated before the middle of RT due to the increasing NIS score that interrupts eating. Capuano et al. (2008) reported that HNC patients develop severe mucositis and dysphagia within 15 days from the start of treatment [23]. 

All HNC patients required ONS during RT in this study. An intensive dietary intervention is recommended before problems arise rather than anticipating decreased intake and lost weight. NIS score will be great tool to identify high-risk patients (i.e., those having severe mucositis and dysphagia) who are most likely benefit from early and aggressive intervention and tube placement. Weight loss ≥5% should be part of the criteria for tube initiation when a low ONS compliance rate is observed among HNC patients. The ESPEN guideline recommends tube feeding in RT-induced severe mucositis or in obstructive tumors resulting in dysphagia, anticipating inadequate energy intake [14]. According to the systemic review, the provision of enteral nutrition support via a tube inserted is suggested once oral intake declines below 50% from the energy requirements or once >5 kg of weight loss has occurred from the baseline [33].

This study has several strengths, which include the use of the validated scored PG-SGA and HNSC^©^ questionnaires. The scored PG-SGA is a valid tool for nutritional assessment in cancer patients. The HNSC^©^ questionnaire has been validated for assessing NIS in HNC. The longitudinal design of this study makes it possible to assess nutritional changes, dietary changes, and the NIS of HNC patients throughout treatment. This is the first study to assess energy and protein intake from ONS aside from dietary energy and protein intake. It is crucial to identify ONS intake in order to provide sufficient energy and protein intake to reduce weight loss. Further studies of effective nutrition intervention programs could be planned earlier in order to identify early symptoms, prevent drastic weight loss and improve treatment outcome. This may highlight the necessity of the continuous screening of nutrition status, dietary changes and NIS to improve nutrition intervention, such as the timely initiation of oral nutritional supplements or enteral tube feeding.

The relatively small sample size may not have been large enough to allow for the generalizability of the results among all HNC patients. Still, this present study has generated preliminary evidence on the current NIS and nutritional and functional statuses of HNC patients serving as a basis for future research. Weight and BMI are not good parameters to evaluate nutrition status due to every HNC patient having unknown cumulative losses during RT. Hence, further studies for nutrition intervention should include NIS monitoring, PG-SGA, dietary intake apart from weight and BMI, and monitoring during treatment. The findings of this study could add to the existing literature on weight changes in HNC during RT, which can help to optimize nutrition intervention.

## 5. Conclusions

In our study, HNC inpatients in the NCI were observed, from the beginning of RT until the end of treatment. The suggested intensive nutritional care time point was the middle of RT, related with a high NIS score, low dietary intake and high ONS intake. This is the first research study that has monitored NIS using the HNSC© checklist to analyze NIS throughout the treatment, and there are several findings that are useful as a contribution for nutrition intervention for HNC patients. ONS compliance affects the percentage of weight loss, muscle mass loss, and total energy and protein intake among HNC patients during RT. As a conclusion, diet monitoring and close advisory has to be conducted for HNC patients in order to ensure better nutrition outcomes during treatment. Further observational studies can be conducted with the NIS checklist to achieve a higher accuracy of ONS compliance for the good of HNC patients’ dietary needs.

## Figures and Tables

**Figure 1 nutrients-12-01225-f001:**
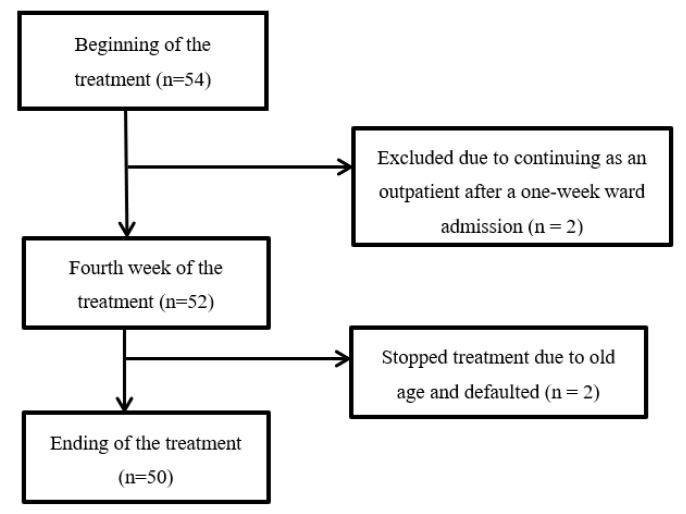
Flow diagram of the protocol.

**Table 1 nutrients-12-01225-t001:** Baseline patient characteristics.

Characteristics	Overall (*n* = 50)	Well nourished (*n* = 22)	Malnourished (*n* = 28)	*p*-Value
**Age (years)**^b^**, median** (IQR)	60 (49–67)	54 (44–67)	61 (52–66)	0.163
**Gender**^c^**: *n* (%)** Male Female	39 (78) 11 (22)	16 (41) 6 (54.5)	23 (59) 5 (45.5)	0.425
**Race**^c^**: *n* (%)** Malay Chinese Indian	21 (42) 19 (38) 10 (20)	9 (42.9) 10 (52.6) 3 (30)	12 (57.1) 9 (47.4) 7 (70)	0.501
**Tumor location: *n* (%)** Tongue Mouth Salivary gland Tonsil Oropharynx Nasopharynx Sinuses Larynx	7 (14) 6 (12) 3 (6) 2 (4) 2 (2) 26 (52) 1 (2) 3 (6)	3 (42.9) 2 (33.3) 1 (33.3) 0 (0) 0 (0) 15 (57.7) 0 (0) 1 (33.3)	4 (58.1) 4 (66.7) 2 (66.7) 2 (100) 2 (100) 11 (42.3) 1 (100) 2 (66.7)	-
**Stage of tumor**^d^**: n (%)** 1–2 3–4	8 (16) 42 (84)	5 (62.5) 17 (34)	3 (37.5) 25 (50)	0.277
**Type of treatment**^c^**: *n* (%)** Radiotherapy Chemoradiotherapy	17 (34) 33 (66)	7 (41.2) 15 (45.5)	10 (58.8) 18 (54.5)	0.773

^b^ Mann Whitney U-test. ^c^ Chi-square test for proportions. ^d^ Fisher’s exact test.

**Table 2 nutrients-12-01225-t002:** Nutritional status, nutrition impact symptoms, ONS compliance, and type of diet at baseline and end of treatment.

Characteristics, *n* (%)	Baseline (*n* = 50)	End (*n* = 50)
**PG-SGA global rating**		
A (well-nourished)	22 (44)	-
B (moderate malnourished)	18 (36)	16 (32)
C (severe malnourished)	10 (20)	34 (68)
**Triage intervention**		
No intervention (Score of 0–1)	4 (8)	-
Health education (Score of 2–3)	13 (26)	-
Dietetic intervention (Score of 4–8)	10 (20)	-
Critical interventions (≥9)	23 (46)	50 (100)
**BMI category** Underweight (<18.5 kg/m^2^) Normal weight (18.5-24.9 kg/m^2^) Overweight (25-29.9 kg/m^2^) Obese (>30 kg/m^2^)	12 (24) 25 (50) 5 (10) 8 (16)	19 (38) 21 (42) 6 (12) 4 (8)
**Weight loss: *n* (%)** None <5% in 1 month 5–10% in 1 month > 10%	14 (28) 30 (60) 5 (10) 1 (2)	1 (2) 15 (30) 19 (38) 15 (30)
Mean ± SD	-	7.39 ± 4.07
**Nutrition Impact Symptoms (NIS)** Yes No	40(80) 10 (20)	50 (100) -
**ONS compliance** Compliant Non-compliant **Type of diet** Normal Soft Liquid diet Tube feeding	- - 36(72) 8 (16) 6(12) -	37 (74) 13 (26) 1(2) 20(40) 23(46) 6(12)

Abbreviations: PG-SGA: patient-generated subjective global assessment, BMI: body mass index, ONS: oral nutrition supplements.

**Table 3 nutrients-12-01225-t003:** Changes in nutritional parameters along the treatment (*n* = 50).

Variables (*n* = 50)	Baseline Mean ± SD	Middle of RT Mean ± SD	End of RT Mean ± SD	X^2^	*p*-Value
Body weight (kg) ^†^	60.24 ± 14.73	57.95 ± 13.92	55.71 ± 13.62	14.731	<0.0001 *
BMI (kg/m2) ^¶^	22.78 ± 5.6	21.92 ± 5.29	21.08 ± 5.19	81.437	<0.0001 *
PGSGA score ^a^	8.72 ± 6.86	-	25.82 ± 5.34	58.47	<0.0001 ^*^
Muscle mass (kg) ^†^	43.03 ± 8.12	41.99 ± 8.28kg	41.30 ± 8.13	4.286	<0.0001 *
Fat mass (kg) ^¶^	15.23 ± 9.17	14.07± 8.43	12.67 ± 8.3	62.23	<0.0001 *
Non-dominant hand handgrip strength (kg) ^†^	25.17 ± 7.46	25.06 ±8.26	24.454 ± 7.97	5.207	0.483
NIS score^¶^	21.78 ± 4.59	39.68 ± 9.54	48.34± 8.79	78.707	<0.0001 *
Total Energy Intake (Kcal/day) ^†^	1412 ± 589	1403 ± 465	1554 ± 482	1.241	0.227
Total protein Intake (g/day) ^†^	63 ± 30	59 ± 22	66 ± 21	3.649	0.312
Total energy intake/current weight (kcal/kg/day) ^†^	23.60 ± 8.58	24.31 ± 8.24	26.26 ± 9.33	1.780	0.270
Total protein intake/ current weight (g/kg/day) ^†^	1.03 ± 0.43	1.02 ± 0.38	1.11 ± 0.36	2.209	0.393
Dietary Energy Intake (kcal/day) ^†^	1355 ± 62	549± 420	413± 426	57.897	<0.0001 *
Dietary protein Intake(g/day) ^†^	60.5 ± 31.29	23 ±23	16 ±21	62.639	<0.0001 *
ONS Energy Intake (kcal/day) ^¶^	56 ±181	854 ±486	1142 ± 572	75.717	<0.0001 *
ONS Protein Intake (g/day) ^¶^	2± 8	35 ±19	49.99 ± 24.42	77.568	<0.0001 *

Mean, standard deviation (SD); * Significant (*p* < 0.05) ^†^ Analyzed by General Linear Model repeated measures. A Greenhouse-Geisser correction for degrees of freedom was used because of deviation from sphericity; ^¶^ Analyzed by the Friedman test; ^a^ Wilcoxon signed rank sum test (baseline vs. end of RT). Abbreviations: PG-SGA: patient-generated subjective global assessment, BMI: body mass index, NIS: nutrition impact symptoms, ONS: oral nutrition supplements.

**Table 4 nutrients-12-01225-t004:** Prevalence of NIS from the HNSC© in participants with severity scores ≥2 for orally fed head and neck cancer patients over time.

Nutrition Impact Symptoms (*n* = 50)	Baseline *n* (%)	Middle of treatment *n* (%)	End of treatment *n* (%)
Taste changes	5 (10)	46 (92)	50 (100)
Difficulty swallowing	9 (18)	42 (84)	47 (94)
Difficulty chewing	26 (52)	43 (86)	47 (94)
Constipation	8 (16)	28 (56)	29 (58)
Loss of appetite	19 (38)	46 (92)	47 (94)
Dry mouth	19 (38)	46 (92)	50 (100)
Pain	11 (22)	38 (76)	45 (90)
Anxious	9 (18)	20 (40)	22 (44)
Nausea	4 (8)	26 (52)	22 (44)
Lack of Energy	14 (28)	40 (80)	46 (92)
Sore mouth	6 (12)	34 (68)	43 (86)
Diarrhea	0 (0)	3 (6)	4 (8)
Thick saliva	12 (24)	44 (88)	48 (96)
Depressed	2 (4)	9 (18)	10 (20)
Fullness	4 (8)	21 (42)	26 (52)
Vomiting	3 (6)	9 (18)	11 (22)
Smell bothersome	6 (12)	32 (64)	36 (72)

Abbreviations: HNSC©, Head and Neck Symptoms Checklist.

**Table 5 nutrients-12-01225-t005:** Weight, muscle, and fat loss, total energy and protein intake, and NIS score according to the ONS compliant group.

Mean ± SD	ONS Compliant Group (*n* = 37)	ONS Non-compliant Group (*n* = 13)	*t*-Value	*p*-Value
Weight loss (%)	6.55 ± 3.65	9.75 ± 4.4	-2.570	0.013 *
Muscle mass loss (kg)	1.34 ± 1.89	2.78 ± 2.61	2.12	0.039 *
Fat mass loss (kg)	2.29 ± 1.73	3.34 ± 2.22	1.74	0.088
Total Energy intake (kcal/day)	1653 ± 462	1273 ± 439	2.579	0.013 *
Total Protein Intake (g/day)	71 ± 19	51 ± 19	3.24	0.002 *
Energy intake (kcal/kg/day)	28 ± 9	22 ± 9	0.079	0.079
Total Protein Intake (g/kg/day)	1.2 ± 0.3	0.9 ± 0.4	0.006	0.006 *
NIS score	47.32 ± 9.1	51.23 ± 7.23	1.391	0.171

Independent T test. * *p* < 0.05 shows a significant difference between the ONS compliant vs. ONS non-compliant groups. Abbreviations: ONS: oral nutrition supplements, NIS: nutrition impact symptoms.

**Table 6 nutrients-12-01225-t006:** Correlation between weight loss percentage and the independent variables (*n* = 50).

Independent Variables	Relationship (r)	Significance (p)
PG-SGA score ^b^	0.600	0.0001 ***
NIS score ^a^ Total Energy intake (kcal/d) ^a^ Total Protein intake (g/d) ^a^	0.363 −0.163 −0.199	0.01 * 0.257 0.167
Diet Energy intake (kcal/d) ^a^ Diet Protein intake (g/d) ^a^	−0.114 −0.127	0.432 0.379
ONS Energy intake (kcal/d) ^b^	−0.049	0.736
ONS Protein intake (g/d) ^b^	−0.034	0.814

^a^ Pearson’s rho test. ^b^ Spearman’s rho test. * *p* < 0.05; *** *p* < 0.001 shows significance. Abbreviations: PG-SGA: patient-generated subjective global assessment, NIS: nutrition impact symptoms, ONS: oral nutrition supplements.

**Table 7 nutrients-12-01225-t007:** Correlation between total NIS score and ONS energy and protein intake (*n* = 50).

	Relationship (r)	Significance (*p*)
**ONS energy intake** Baseline	0.170	0.239
Middle of RT	0.429	0.002 *
End of RT	0.106	0.466
**ONS protein intake** Baseline	0.159	0.269
Middle of RT	0.412	0.003 *
End of RT	0.124	0.389

* Significant with *p* <0.05 Spearmen’s rho. Abbreviations: ONS: oral nutrition supplements.

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
