# Peer review of "Changes in Nutrition Impact Symptoms, Nutritional and Functional Status during Head and Neck Cancer Treatment"

_nutrients, 2020, doi:10.3390/nu12051225_

Round 1

Reviewer 1 Report

Dear authors,

Thank you for putting together this article, which is helpful for physicians and nutritionists supporting head and neck cancer patients.

I have several comments that I hope will improve the value of your article.

First of all, it's not clear when reading the article at what time points measurements were taken. It seems like there were measurements taken before treatment, at 4 weeks and then at the end of treatment. It would be helpful to describe the measurement time points in the methods section, and provide information on how long the patients undergo treatment.

The NIS provides a list of symptoms that patients are suffering from. It would be useful to determine whether particular symptoms that are measured are associated with both weight loss and non-compliance to the ONS. This could help in improving compliance, or help identify whether the weight loss is due to non-compliance, or more severe symptoms that cause both weight loss and non-compliance.

Specific comments in the text:

FIGURE 1: include the N for subjects excluded/leaving the study
TABLE 1: why was no p-value calculated for the race and tumour location categories?
376-377: did patients experience dry mouth due to treatment?

Author Response

Point 1: First of all, it's not clear when reading the article at what time points measurements were taken. It seems like there were measurements taken before treatment, at 4 weeks and then at the end of treatment. It would be helpful to describe the measurement time points in the methods section, and provide information on how long the patients undergo treatment.

Response 1: Page 2, line 112-114. There were three measurement points in this study to assess NIS, nutritional status and functional status included at the baseline (week 1), middle (week 4) and the end of radiotherapy (week 7). Page 2, line 105-106. HNC received RT dosage between 60Gy to 70Gy in daily factions of 2.0 Gy within 7 weeks while CCRT patients were received additional of weekly cisplatin or carboplatin during the 7 weeks of RT.

Point 2: The NIS provides a list of symptoms that patients are suffering from. It would be useful to determine whether particular symptoms that are measured are associated with both weight loss and non-compliance to the ONS. This could help in improving compliance, or help identify whether the weight loss is due to non-compliance, or more severe symptoms that cause both weight loss and non-compliance.

Response 2: Page 8, line 332-333. Table 4. Prevalence of NIS from the HNSC© in participants with severity scores ≥2 for orally-fed head and neck cancer patients over time showed the results of prevalence of NIS at the end of treatment. The association between particular symptoms with weight loss and non-compliance to the ONS will be consider for further analysis as the current objective for this paper is to analysis changes of NIS, nutrition status and functional status by compare the results between end of radiotherapy with baseline and the trends during treatment.

Point 3: Specific comments in the text:              Response 3:

FIGURE 1: include the N for subjects excluded/leaving the study

Page 5, line 263 (n=2)

Page 5, line 267 (n=2)

TABLE 1: why was no p-value calculated for the race and tumour location categories?

Page 7, line 284. More than 20% of all cells have an expected frequency < 5

376-377: did patients experience dry mouth due to treatment?

Page 9, line 331. Yes, all patients experienced taste changes and dry mouth at the end of treatment.

Reviewer 2 Report

 1) Why did you use non-dominant hand for evaluating handgrip? You said at discussion that handgrip strength may not be of benefit in the nutritional assessment of these patients and should not be part of routine assessment. Nevertheless, the main issue of measuring muscular mass is that it usually has a good correlation to a decrease in patient functional capacity. In most of pathologies, a decrease in handgrip strength reveals worst prognostic and death, so I think you don't have to be so emphatic on that. 

2) It is not described the incidence of severe mucositis and dysphagia. At conclusions, you emphasize the diets importance on patient's nutritional status (and that's true), but in your study, bad nutritional results were reached with diets counseling and survey... I wonder if many of this patients could need early feeding tube placement, or at least in a higher proportion?

Author Response

Point 1:  Why did you use non-dominant hand for evaluating handgrip? You said at discussion that handgrip strength may not be of benefit in the nutritional assessment of these patients and should not be part of routine assessment. Nevertheless, the main issue of measuring muscular mass is that it usually has a good correlation to a decrease in patient functional capacity. In most of pathologies, a decrease in handgrip strength reveals worst prognostic and death, so I think you don't have to be so emphatic on that.

Response 1: In this study, non-dominant hand was chosen for evaluating handgrip strength, according to Lakenman et al. 2017, which proposed non-dominant hand to measure handgrip strength during CRT among esophageal cancer patients. We have removed that point in discussion.

Point 2:  It is not described the incidence of severe mucositis and dysphagia. At conclusions, you emphasize the diets importance on patient's nutritional status (and that's true), but in your study, bad nutritional results were reached with diets counseling and survey... I wonder if many of this patients could need early feeding tube placement, or at least in a higher proportion?

Response 2: Page 8, line 332-333. Table 4. Prevalence of NIS from the HNSC© in participants with severity scores ≥2 for orally-fed head and neck cancer patients over time showed the results of prevalence of NIS at the end of treatment.

Page 13, line 500-507. In this study, the ONS compliant group had >5% critical weight loss despite dietary counselling and ONS support. The average ONS compliant group achieved 28 ± 9 kcal/kg, which was below 30 kcal/kg/day when compared with the current recommendations. The standard compliance rate in this study, according to Hopanci’s study was set at ≥75% as a benchmark for compliance. However, the ONS compliant group in this study still experience critical weight loss which means that 75% could not reach the standard compliant rate for HNC patients undergoing RT. Perhaps a suggestion of a 100% compliance rate only can consider compliance for better nutrition outcome monitoring.

Page 14, line 532. Weight loss ≥5% should be part of the criteria for tube initiation when a low ONS compliance rate is observed among HNC patients. The ESPEN guideline recommend tube feeding in RT-induced severe mucositis or in obstructive tumors resulting in dysphagia, anticipating inadequate energy intake. According to the systemic review, the provision of enteral nutrition support via a tube inserted is suggested once oral intake declines below 50% from the energy requirements or >5 kg of weight loss has occurred from the baseline.

Reviewer 3 Report

With this study, the authors examined the changes in nutrition impact symptoms (NIS), nutritional and functional status, occurring throughout the radiotherapy in head and neck cancer patients. The paper would be of interest, but I have several concerns about its publication and I would like to reconsider it after a major revision:

  1. There are many structural errors. The manuscript’ structure should definitely be improved.
  2. The introduction does not really guide the reader to the main hypothesis and the work novelty.
  3. The reader is somewhat lost right from the start because of the lack of a guiding framework that helps with the evaluation of the information that is presented.
  4. The sample size looks small. Please calculate the exact power.
  5. Weight and BMI are not good parameters to evaluate the nutritional status. This aspect should be discussed as limitation.
  6. An analysis by gender would be interesting.
  7. Discussion needs a major improvement. Limitations and strength should be added.

Round 2

Reviewer 3 Report

Article has serious flaws and revision is incomplete. A power of 0.8 is too low and the analysis of a gender effect is important for this kind of results. The article remains confusing and difficult for a reader to interprete novelty and clinical importance.